# Early Myocardial Strain Reduction and miR-122-5p Elevation Associated with Interstitial Fibrosis in Anthracycline-Induced Cardiotoxicity

**DOI:** 10.3390/biomedicines13010045

**Published:** 2024-12-27

**Authors:** María de Regla Caballero-Valderrama, Elisa Bevilacqua, Miriam Echevarría, Francisco Javier Salvador-Bofill, Antonio Ordóñez, José Eduardo López-Haldón, Tarik Smani, Eva M. Calderón-Sánchez

**Affiliations:** 1Cardiology Unit, University Hospital Virgen del Rocío, 41013 Seville, Spain; caballerovmr@gmail.com (M.d.R.C.-V.); jelhaldon@gmail.com (J.E.L.-H.); 2Cardiovascular Pathophysiology Group, Institute of Biomedicine of Seville-IBiS, University of Seville/Hospital Universitario Virgen de Rocio/CSIC, 41013 Seville, Spain; elisa.bevilacqua.eb@gmail.com (E.B.); antorfernan@us.es (A.O.); tasmani@us.es (T.S.); 3Department of Medical Physiology and Biophysics, School of Medicine, University of Seville, 41009 Seville, Spain; irusta@us.es; 4Oncology Unit, University Hospital Virgen del Rocío, 41013 Seville, Spain; franciscoj.salvador.sspa@juntadeandalucia.es

**Keywords:** cardiotoxicity, anthracycline, epirubicin, global longitudinal strain, circumferential strain, myocardial fibrosis, edema, aquoporin-1, array, miR-122-5p

## Abstract

Echocardiographic myocardial strain is crucial for early detection of anthracycline-induced cardiotoxicity, particularly in patients at moderate or high risk. Background/Objectives: This study investigates changes in global longitudinal strain (GLS) in breast cancer patients with low baseline risk for cardiotoxicity during cancer therapy. We also examined the relationship between echocardiographic strain, structural myocardial changes, and microRNA (miRNA) dysregulation associated with cancer treatment using an animal model. Methods: Echocardiography and blood tests were examined in 33 breast cancer patients with low baseline risk for cardiotoxicity during anthracycline treatment, with a follow-up at 12 months. Additionally, 16 Wistar rats received epirubicin (20 mg/kg over 4 weeks) to examine cardiac strain and structural changes. Moreover, circulating miRNA levels were assessed in patients’ serum using microarray at the end of the treatment and further analyzed in peripheral blood from the animal model. Results: Pathological GLS values were observed in 27.27% of patients after four cycles, with 15.15% showing reduced left ventricular ejection fraction (LVEF) after 12 months. In the animal model, epirubicin-induced circumferential strain (CS) decrease correlates with myocardial fibrosis assessed histologically and by a significant increase in *COL1* and *TGFB2* expression. Furthermore, we found a significant decrease in aquaporin1 expression associated with the presence of vacuoles in treated rats. Furthermore, dysregulation in the expression of miRNAs was observed in patients with cardiotoxicity. Among them, hsa-miR-122-5p is increased in both patient and rat serum post-treatment. Conclusions: A notable percentage of low-risk patients exhibited cardiac strain reduction due to cardiotoxicity. Epirubicin treatment caused structural heart changes in rats, highlighting miR-122-5p as a potential fibrosis marker that correlated with echocardiographic parameters.

## 1. Introduction

Cardiovascular disease and cancer are the most prevalent health problems and the main causes of death worldwide [1]. Fortunately, improvement in early detection techniques and treatments has caused an increase in survival. However, in patients who survive cancer, cardiovascular diseases are, along with second neoplasms, a major cause of death [2]. It is also known that certain cancer treatments triple the risk of cardiovascular diseases [3]. Therefore, in these patients the objective is not only to cure the disease, but also to early detect and treat the complications derived from cancer treatment. Cardiotoxicity due to cancer treatment is defined as the occurrence of any cardiovascular complication related to it. Among such complications, cancer-therapy-related cardiac dysfunction (CTRCD) is one of the most important types of cardiotoxicity because of its prevalence, morbidity, and mortality [2]. However, not all the patients have the same risk of developing cardiotoxicity. Depending on some factors, the risk of cardiotoxicity can be classified as low, moderate, or high [4]. Specifically for anthracyclines, this risk is based on the presence of previous cardiovascular disease (heart failure, myocardial disease, cardiotoxicity, severe valvular diseases, ischemic cardiomyopathy, low ejection fraction), rise in cardiac biomarkers, cardiovascular risk factors (age, arterial hypertension, diabetes mellitus, chronic kidney disease, obesity or smoking), previous oncological diseases with previous anthracycline or radiotherapy treatment, or if the patient is going to receive another cardiotoxic agent as anti-HER2 treatments. These risk factors are taken into account when calculating the risk of cardiotoxicity and, depending on this risk, the follow-up of these patients is protocolized according to guidelines [2], both with echocardiography and with cardiac biomarkers [2,5,6], although most of the studies have been performed in patients at moderate or high risk, and the evidence is scarce for patients at low risk for cardiotoxicity.

Anthracyclines are chemotherapeutic drugs that are part of the standard treatment in breast cancer, and they are associated with the development of CTRCD [7]. It is well defined that its development is dose-dependent and, when CTRCD is early detected and treated, it can be reversible [8]. Anthracyclines’ mechanism of action is based on topoisomerases block, leading to cardiac cell death, cytokines release like transforming growth factor β (TGF-β), fibroblast activation, increased collagen deposition, and alteration of different metalloproteinases resulting in interstitial fibrosis [9,10]. In addition, vacuolar changes in myocytes and interstitial edema have been found in the hearts of large animals treated with anthracyclines [11,12]. This intracellular vacuolar degeneration has been described as the earliest marker of anthracycline-induced cardiotoxicity, and it has been correlated with abnormal values in T2-mapping in Cardiac Magnetic Resonance (CMR) [11]. However, the implicated mechanism in anthracycline-induced heart edema has not been described yet.

Different biomarkers have been proposed for the anthracyclines CTRCD early detection (NT-pro-BNP, troponins, vasopressin, sST2, galectin 3, etc.) [13,14,15]. MiRNAs have been proposed as predictive markers of different pathologies related to heart damage [16]. MiRNAs are small single-stranded non-coding RNAs that act mostly as negative regulators of gene expression by complementary base pairing in the 3′-region of mRNA (3′-UTRs) [17]. Different miRNAs have been suggested as players in anthracycline damage, such as *miR-208a* [18] and *miR-146a* [19].

Herein, we analyzed a small cohort of breast cancer patients treated with anthracyclines and with a low-risk profile for cardiotoxicity, in which we described how left ventricular strain and ejection fraction parameters evaluated by echocardiography are affected. Furthermore, we studied the different pattern in fibrosis and related miRNAs in patients with and without cardiotoxicity. Then, we designed an animal model of cardiotoxicity in which we studied the left ventricular strain and ejection fraction, myocardial fibrosis, edema and, finally, the relationship between strain and structural changes in cardiac tissue and circulating miRNAs.

## 2. Materials and Methods

### 2.1. Study Setting and Approvals

This study was performed in accordance with the recommendations of the Royal Decree 53/2013 in agreement with Directive 2010/63/EU of the European Parliament and approved by the local Ethics Committee on Human Research of the Virgen del Rocío University Hospital in Seville (single-center study with permission number: 0987-N-18; date of approval: 5 July 2018; all the patients signed the informed consent), and the Animal Research Committee of the Andalusian Government (Consejería de Agricultura y Pesca; permission number: 05/09/2018/135; date of approval: 11 September 2018).

### 2.2. Study Population

Female patients over 18 years old with newly diagnosed early stage HER2 negative breast cancer, candidates for adjuvant or neoadjuvant anthracycline treatment, were consecutively recruited at our hospital for one year. The exclusion criteria were to have previous cardiac, cerebrovascular, or peripheral arterial diseases, to have more than one cardiovascular risk factor even with adequate control under treatment, to be on treatment with beta-blockers and angiotensin-converting enzyme inhibitors (ACEI), to have had a previous onco-hematological disease treated with chemotherapy or radiotherapy, to have metastatic breast cancer, to have a LVEF lower than 53% at baseline echocardiography, or to have an inadequate image quality for GLS and 3D-LVEF quantification. With these criteria, in accordance with the most recent scales [4], we recruited patients at low baseline risk for cardiotoxicity.

Patients received four cycles of anthracycline (one every 21 days, Appendix A). Echocardiography was performed at baseline and after cycles two and four of anthracyclines (CYCLE 2 and CYCLE 4, respectively) (between three and five days after the cycle), as well as 12 months after completing this treatment. Cardiotoxicity was defined as a decrease in LVEF > 10% compared with baseline value, with final LVEF < 53%, allowing us to consider two groups: CTRCD-group and non-CTRCD-group. Once cardiotoxicity was detected, cardioprotective therapies with beta-blockers and ACEI were started and maintained during the follow-up.

### 2.3. Animal Model

The animal model was conducted in the animal facilities of the Institute of Biomedicine of Seville. To study heart alterations induced by the treatment with anthracyclines, and to stablish the animal model, we used different cumulative doses of epirubicin: control (saline), 10 mg/kg, 20 mg/kg and 36 mg/kg applied in four weeks. The lethal dose for rats was previously determined at 14.57 mg/kg/dose [20]. We included 3–5 rats in each group (weighting around 250 g, 8–12 weeks old). We evaluated death rate, interstitial fibrosis, and decrease in LVEF versus control group (Appendix A); according to these data, we decided to use the cumulative dose of 20 mg/kg applied at 5 mg/kg/dose during four weeks for the rest of experiments.

A total of 16 healthy female Wistar rats were used as the animal model: 8 were treated with epirubicin (treatment group), and 8 were in the control group (Appendix A). Rats from the treatment group received intraperitoneal epirubicin cycles. In the animal model, cardiotoxicity was considered when we observed changes in LVEF over 15% in the treated group. Echocardiography was performed at baseline, 24 h after the second epirubicin or saline dose, and finally, 24 h after the last dose (CYCLE 2 ECHO and CYCLE 4 ECHO, respectively). Serum samples were obtained from each animal at baseline and before each echocardiography. Twenty-four hours after the fourth dose, animals were euthanized by intraperitoneal administration of a lethal dose of sodium thiopental (200 mg/kg) for tissue and blood sample collection, as shown in Appendix A.

### 2.4. Transthoracic Echocardiograms

All studies in patients were acquired with a Philips iE33 ultrasound system with a X5-1 transducer (Philips Health Care, Andover, MA, USA). Echocardiographic parameters included 3D-LVEF and GLS, both were quantified using QLAB 10.7 (Philips Health Care, Santa Barbara, CA, USA). A relative fall of more than 15% or an absolute fall above −19% in GLS were considered pathological.

Animals were anesthetized with sevoflurane as previously described [21]. Transthoracic echocardiograms were acquired with a General Electric Vivid i ultrasound system with a 10 MHz transducer (General Electric Healthcare, Chicago, IL, USA). Echocardiographic parameters included LVEF (quantified by monoplane Simpson), radial strain (RS) and circumferential strain (CS). EchoPAC™ v110.1.2 software (General Electric Healthcare, Chicago, IL, USA) was used to quantify these parameters.

### 2.5. Study of Fibrosis and Edema in Rat Heart Tissue

Rat hearts were fixed with formalin and followed the dehydration protocol. Hearts were included in paraffin and cut at the microtome with a section of 6 μm; sections were stained following Masson’s trichrome protocol to detect fibrosis and edema. Images were acquired with an Olympus BX-61 (Olympus, Tokyo, Japan) direct fluorescent microscope at 40× objective in bright field, and blue pixels were analyzed with CellProfiler software (http://www.cellprofiler.org). Edema was observed qualitatively as extracellular vacuoles in histological pictures.

### 2.6. Genes qRT-PCR

Total RNA was extracted from rat heart tissues by homogenization using TissueLyzer II (Qiagen, Hilden, Germany). RNA was obtained with the miRNeasy Mini Kit (1038703, Qiagen, Germany) according to the manufacturer’s instructions. To perform qRT-PCR, RNA was retrotranscribed to cDNA with iScript cDNA Synthesis Kit (1708891, Biorad, Hercules, CA, USA), and data analysis was performed with QuantStudio™ Real-Time PCR software (version 1.2, Thermofisher, Waltham, MA, USA). Fold-change quantification was calculated using the comparative cycle threshold CT (ΔCT) method, normalized to *18S* gene. The studied genes were transforming grown factor beta 1 and 2 (*TGFB1* Forward: 5′-ATTCCTGGCGTTACCTTGG-3′; Reverse: 5′-AGCCCTGTATTCCGTCTCCT-3′; *TGFB2* Forward: 5′-GCAGAGTTCAGGGTCTTTCG-3′; Reverse: 5′-GCTGGGTTGGAGATGTTAGG-3′), collagen I and III (*COL1* Forward: 5′-TTCACCTACAGCACGCTTGT-3′; Reverse: 5′-TTTGGGATGGAGGGAGTTTA-3′; *COL3* Forward: 5′-GGTCACTTTCACTGGTTGACGA-3′; Reverse: 5′-TTGAATATCAAACACGCAAGGC-3′) (Sigma, Livonia, MI, USA).

### 2.7. Immunofluorescence

Wheat Germ Agglutinin (WGA) staining was used to assess cardiomyocyte cross-sectional area as previously described [22]. Hearts of rats were fixed with 4% paraformaldehyde, then included in OCT (Sakura, Tokyo, Japan) and frozen. Hearts were cut in 6 µm slices and stained with WGA-Alexa Fluor 488 (1:100, W11261, Thermofisher, Waltham, MA, USA) and antiaquaporin 1 antibody (1:50, sc-25287, Santa Cruz, Dallas, TX, USA). Images were acquired with fluorescence microscope Olympus BX61 (Olympus, Tokyo, Japan) and analyzed with ImageJ software (version 1.45, National Institute of Health, Bethesda, MA, USA).

### 2.8. Western Blotting

Protein samples were extracted from rat hearts after CYCLE 4. A total of 40 μg of protein lysate was subjected to SDS-PAGE gel at 10% acrylamide and electrotransferred onto PVDF membranes. Membranes were incubated overnight at 4 °C with specific primary antibodies in Tris-buffered saline containing 0.1% Tween 20 (TTBS) with 1% of BSA: α-SMA (1:500, A5228, Sigma, Livonia, MI, USA), MMP1 (1:1000, GT2649, Gene Tex, Irvine, CA, USA), AQP1 (1:500sc-25287, Santa Cruz, Dallas, TX, USA). Detection was performed with the enhanced chemiluminescence reagent Clarity Western ECL Substrate (Biorad, Hercules, CA, USA) using Chemidoc (Thermofisher, Waltham, MA, USA). The quantification was performed at ImageLab (Biorad, Hercules, CA, USA). GAPDH (1:10,000, G9545, Sigma, Livonia, MI, USA) was used as a housekeeping control.

### 2.9. microRNAs Array

Total RNA was extracted from peripheral blood serum with miRNAeasy mini kit (Qiagen, Hilden, Germany) according to the manufacturer’s instructions. Arrays was performed from patients’ serum after CYCLE 4 non-CTRCD-group (n = 5) and in CTRCD-group (n = 5). The total cDNA was labeled using the FlashTag^®^ Biotin HSR labeling Kit (Thermofisher, Waltham, MA, USA) following instructions supplied in the user manual. GeneChip^®^ miRNA 4.0 arrays (Thermofisher, Waltham, MA, USA) were used for miRNAs analysis, with a superior performance: 0.95 of reproducibility (inter- and intra-lot), >80% of transcripts detected at 1.3 nmol from 130 ng of total RNA, 4 logs of dynamic range and >0.97 signal correlation > 0.94 fold-change correlation. Washing, staining (GeneChip^®^ Fluidics Station 450, Thermofisher, Waltham, MA, USA), and scanning (GeneChip^®^ Scanner 3000, Thermofisher Scientific, Waltham, MA, USA) were performed following manufacturer’s protocol. Briefly, importing the CEL file, the analysis of miRNA level RMA + DABG-All and exporting of the results were performed using Transcriptome Analysis Console (TAC) 4.0 software (Thermofisher, Waltham, MA, USA). A comparative analysis between non-CTRCD-group and in CTRCD-group was carried out using fold-change of over ±2.0 with a *p*-value < 0.05. Age was applied as a real covariate in the in silico analysis with TAC software.

### 2.10. In Silico Analysis

In silico analysis of possible microRNAs regulating apoptosis and fibrosis related genes was performed with miRDB (www.mirdb.org, accessed on 2 February 2024) and TargetScan (www.targetscan.org, accessed on 2 February 2024) to evaluate possible target genes. Later, the target genes were filtered with GeneVenn (http://genevenn.sourceforge.net/, accessed on 2 February 2024). Target genes’ possible implications in different biological pathways were studied with Panther Gene (https://www.pantherdb.org/, accessed on 2 February 2024).

### 2.11. microRNAs RT-qPCR

To perform qRT-PCR, RNA was retrotranscribed to cDNA with miRCURY LNA RT Kit (Qiagen, Germany) for microRNAs detection. Primer PCR mix for microRNAs included miRCURY LNA SYBR Green PCR kit (Qiagen, Germany) and *hsa-miR-122-5p* miRCURY LNA miRNA PCR Assay. Data analysis was performed with QuantStudio™ Real-Time PCR Software (Thermofisher, Waltham, MA, USA). Fold-change quantification was calculated using the comparative cycle threshold (CT) method, using *hsa-miR-26a* as endogenous control. Each sample was analyzed in triplicate, and samples presenting a Ct value > 35 and a Ct confidence value ≤ 0.6 were excluded.

### 2.12. Statistical Analysis

Quantitative variables following a normal distribution are presented as mean ± standard deviation (SD) and categorical variables as absolute and relative frequencies (%). Shapiro–Wilk test was used to study data distribution. ANOVA test was used to study the echocardiographic parameters evolution of the whole patient cohort. Differences between patients from CTRCD-group and non-CTRCD-group were determined by Student’s *t*-test for quantitative variables and Chi2 test for categorical variables (Fisher test in case of n < 5). Repeated measures ANOVA test was used to study the behavior of GLS between groups of CTRCD and non-CTRCD. For the animal study, ANOVA test was used to analyze echocardiographic parameters evolution of the whole cohort and repeated measures ANOVA to study the evolution of CS and RS in rats developing cardiotoxicity compared with those who did not. Differences in fibrosis between treatment and control groups were determined by Student’s *t*-test. Pearson’s correlation test was performed between fibrosis and edema variables, CS, GLS, and miRNAs. Statistical analysis was performed using SPSS Statistics (version 26, IBM corp., Armonk, NY, USA), and GraphPad Prism software (version 9, Boston, MA, USA) was used for figures. An α limit under 0.050 was considered statistically significant.

## 3. Results

### 3.1. Development of CTRCD in the Clinical Cohort Assessed by Changes in GLS

A total of 40 patients were included in the study, and 33 of them completed the follow-up (82.5%). In five of the seven excluded patients, the reason was the development of metastasis, and in the other two, it was a poor acoustic window due to left mastectomy. All participants were women with a mean age of 52.8 ± 10.2 and were considered at low risk according to the most recent scales [2,4]. Baseline data and treatment details of the whole cohort broken down by the presence of CTRCD are shown in Table 1.

As shown in Table 1 and in Appendix A, during the 12 months of follow-up after completing the epirubicin treatment, five patients (15.15%) developed CTRCD, defined as a decrease in LVEF > 10%, with final LVEF < 53%. Two of them were symptomatic, experiencing dyspnea, while three were asymptomatic. All patients received treatment with beta-blockers and ACEI upon CTRCD diagnosis. Changes over time in GLS and LVEF are shown in Table 2.

In the whole cohort, GLS was significantly decreased from baseline (−21.73 ± 1.74%) to CYCLE 4 ECHO value (−20.28 ± 1.78%; *p* = 0.011) (Table 2). When studying the evolution of GLS values in CTRCD-group and non-CTRCD-group over time, we observed significant differences from ANOVA analysis (*p* < 0.001, Figure 1A), specifically at CYCLE 4 ECHO (*p* = 0.039, Figure 1B). Figure 1B shows significant differences in the CTRCD group between baseline (−22.94 ± 1.46%) and CYCLE 4 ECHO value (−18.32 ± 1.40%) (*p* < 0.0001), and between CYCLE 2 ECHO (−21.60 ± 1.06%) and CYCLE 4 ECHO values (*p* = 0.002). No differences were observed in the non-CTRCD group over time. These pathological GLS values, defined as a relative fall of more than 15% or an absolute fall above −19%, were observed in nine patients (27.27%) after the fourth anthracycline cycle. All the patients who later developed CTRCD (n = 5; 15.15%) had previously presented a pathological decrease in GLS at CYCLE 4 ECHO.

### 3.2. Changes in Echocardiographic Parameters in the Animal Model

For the whole treatment group of rats, we observed a significant decrease in CS, RS, and LVEF over time, as shown in Table 3. We detected a pathological worsening in LVEF in five of the eight rats at CYCLE 4 ECHO (69.00 ± 4.35%), allowing us to consider two groups: CTRCD group (n = 5) and non-CTRCD group (n = 3). There were no changes in the control group.

When studying CS in these two groups over time, we found globally significant differences (*p* = 0.045) as observed in Figure 2A. Figure 2B also shows a significant decrease in CS in the CTRCD group from baseline (−23.88 ± 2.84%) to CYCLE 2 value (−17.19 ± 4.47%) (*p* = 0.003) and to CYCLE 4 value (−15.69 ± 1.38) (*p* < 0.001). In case of RS, we did not find global differences between groups (Figure 2C), but, as shown in Figure 2D, there were significant differences in CYCLE 4 ECHO value between CTRCD group (30.62 ± 7.31%) and non-CTRCD group (47.11 ± 4.17) (*p* = 0.011). As summarized in Figure 2D, there was a significant decrease in RS in the CTRCD group from baseline (49.84 ± 7.98%) to CYCLE 2 ECHO value (38.88 ± 5.45) (*p* = 0.029) and to CYCLE 4 ECHO value (30.62 ± 7.31%) (*p* < 0.001). There were no changes over time for CS and RS in non-CTRCD group.

### 3.3. Myocardial Fibrosis Is Associated with CS Decrease

We used the rat animal model of cardiotoxicity to assess fibrosis using staining with Masson trichrome in sections of rat hearts. The presence of fibrosis in the heart parenchyma was not detected in the control group, as shown in the representative image in Figure 3A; meanwhile, Figure 3B shows consistent staining, indicating an increase in the expression of collagen fibers in the treated group, as also quantified in Figure 3C (*p* = 0.020). Moreover, we observed a significantly higher gene expression of *COL1* (*p* = 0.013) and *TGFB2* (*p* = 0.021) in cardiac tissue of the treatment group as compared with the control group (Figure 3D,E). In contrast, there were no differences in the expression of *COL3* or *TGFB1* (Appendix A). The treatment group also showed a lower expression of MMP1 protein (Figure 3F, *p* = 0.020), but the expression of α-SMA protein tended to increase (Figure 3G, *p* = 0.064) when compared with the control group. Finally, we observed a strong correlation between the decrease in left ventricular CS with myocardial fibrosis assessed histologically (Figure 3H; r = 0.899, *p* = 0.002) and with the expression of *COL1* (Figure 3I; r = 0.739, *p* = 0.036) and *TGFB2* (Figure 3J; r = 0.9, *p* = 0.002).

### 3.4. Edema Detection in Rat Myocardium After Epirubicin Treatment

Histological observations revealed alterations in myocardial tissue beyond fibrosis. For instance, Figure 4A shows the presence of small round bubbles between myocardial fibers in treated animals, indicating qualitatively edema. Therefore, we evaluated the level of AQP1 protein, a key player in water homeostasis. As shown in Figure 4B, hearts from treated animals had significantly decreased expression of AQP1 as compared with the controls (*p* = 0.020). Immunofluorescence of AQP1 and WGA was performed. Control animals showed a high staining of AQP1 (Figure 4C), while rats from the treatment group showed less staining of AQP1 (Figure 4C). The analysis of cardiomyocyte cross-sectional area quantified using WGA indicates a decreased tendency in cell size in treated rats, as shown in Figure 4D and summarized in Figure 4E.

### 3.5. miRNAs Expression in Blood Serum of Patients and Its Validation in Rat Model

To further analyze possible regulation of genes associated with fibrosis and edema after epirubicin treatment, we performed an array of miRNAs in the serum of patients collected from non-CTRCD-group (n = 5) and in CTRCD-group (n = 5) after CYCLE 4. Figure 5A confirms that data from the two groups are well separated in the principal component analysis (PCA) control, confirming the quality of these samples. As shown in Figure 5B, we found 47 differently expressed miRNAs, particularly, 17 were downregulated and 30 upregulated between the two groups (Table 4). Data in Figure 5C,D show the volcano plot (Figure 5C) and the hierarchical cluster (Figure 5D): patient data from each group that appeared well clustered. Within those miRNAs, has-miR-122-5p exhibits the most considerable fold-change and significant *p*-value between both groups.

Furthermore, in silico analysis showed that has-miR-122-5p is predicted to regulate the expression of many genes related to apoptotic and fibrotic processes, such as the P53 pathway, TGF-beta, and FGF signaling, as shown in Figure 6A and Appendix A. Therefore, the expression of miR-122 was examined by qRT-PCR of the serum of the rest of the patients. Figure 6B confirms that miR-122-5p was enhanced at CYCLE 4 in the CTRCD-group, as compared with the basal level of this group. By contrast, there were no differences in its expression between the basal and CYCLE 4 in the non-CTRCD group. When miR-122 expression at CYCLE 4 was normalized to the basal level of each patient, we observed an almost significant increase in its level in the CTRCD group as compared with non-CTRCD (Figure 6C).

In order to analyze whether the same miRNA is sensitive to epirubicin in the animal model, we assessed the expression of miR-122-5p in rats treated with epirubicin. Figure 6D shows that miR-122-5p was significantly elevated after CYCLE 4 in the animals’ serum in comparison with its basal level. Moreover, in Figure 6E, we observed a negative correlation between changes in the expression of miR-122-5p and the LVEF decrease (*p* < 0.0478; r = −0.9522, Figure 6E), although there was no significant correlation between miR-122-5p expression and GSL fall in patients (Appendix A). All together, these data indicate that the expression of miR-122-5p is sensitive to epirubicin treatment, both in patients and the animal model.

## 4. Discussion

Nowadays, it is well known that the early detection of cardiotoxicity due to cancer therapies is crucial to avoid the progress of the disease toward heart failure [2,4,23]. In this study, in a 33-patient cohort with breast cancer and a low baseline risk for cardiotoxicity treated with anthracyclines, we have described significant changes in GLS and LVEF during anthracycline treatment. Even if our study population had a low-risk profile for cardiotoxicity, an important percentage of some of these patients still had cardiotoxicity associated with LVEF fall. Additionally, we could reproduce anthracycline-induced cardiotoxicity in a Wistar rat model, and we observed structural changes, such as myocardial fibrosis and interstitial edema, together with an overexpression in fibrotic genes, showing a strong correlation between fibrosis parameters and the decrease in CS. Moreover, we observed increased levels in peripheral blood of miR-122-5p both in humans and rats, suggesting its possible relationship with the observed fibrosis.

Until the recent publications of the first cardio-oncology guidelines, the most accepted definition for cardiotoxicity was a reduction of more than 10% in left ventricular ejection fraction (LVEF), compared with the baseline value, with a final LVEF value of less than 53% [4,5]. Currently, its diagnosis focuses on the development of heart failure or the decrease in LVEF, the decline in global longitudinal strain (GLS) [2], or the rise in cardiac biomarkers (cardiac troponin or natriuretic peptides) [2]. Depending on some factors, such as the presence of cardiovascular risk factors or the history of previous cardiovascular disease or oncological disease, the risk of cardiotoxicity can be classified as low, moderate, or high [4]. Our study focused on HER2-negative breast cancer patients treated with anthracyclines who had a low baseline risk for cardiotoxicity, according to the most recent scales. We have provided information about cardiovascular risk factors in Table 1, offering a preliminary overview of the patient profiles in our small cohort. Recruiting patients with homogeneous criteria to be considered only at low risk for cardiotoxicity between 2018 and 2019 was quite difficult, resulting in a small cohort size, which limited the possibility of performing advanced statistical studies. For this reason, we only performed a descriptive study with our clinical cohort. We observed a pathological worsening in GLS, considering both relative and absolute falls, after the fourth cycle of anthracycline treatment in 27.27% of the patients, which agrees with the published literature [23,24]. In our cohort with breast cancer patients and low cardiovascular risk, we observed that 15.15% of patients developed CTRCD. Moreover, a recent study used CMR to analyze cardiotoxicity in a low-risk cohort of 59 female patients with breast cancer treated with anthracycline. They found a larger percentage of CTRCD (54.2%) after 12 months, defined as a reduction in LVEF and strain fall as indicators of CTRCD [25]. Perhaps, using CMR, which is a more sensitive imaging technique, allowed such an increase in the detection of cardiotoxicity. These findings, in accordance with our data, confirmed the lack of conclusive evidence regarding the appropriate timing for echocardiography in the follow-up of patients with a low baseline risk for cardiotoxicity because most of the published studies have focused on patients with moderate or high risk for cardiotoxicity [2,5,6].

In order to evaluate the damage afforded by anthracycline treatment on the heart, we used a rat model for cardiotoxicity. We demonstrated that epirubicin induced a significant decrease in CS before the decrease in LVEF in healthy rats, in a similar way as in humans. We found that chemotherapy with epirubicin in rats evoked an interstitial myocardial fibrosis with a significantly higher expression of *COL1* and *TGFB2*. In contrast, we did not observe any differences in the expression of *COL3* or *TGFB1,* which is in agreement with previously described data [26]. Additionally, αSMA protein trended to increase, suggesting fibroblast activation, as previously described in ischemic cardiomyopathy [27] and in cardiotoxicity [28]. We also observed a significant downregulation of MMP1 at the protein level. By contrast, other studies suggested an increase in *MMP1* mRNA expression, but not at protein levels after chemotherapy [26]. MMP1 has been proposed as a modulator of extracellular matrix homeostasis that is involved in Collagen I degradation in the extracellular left ventricle matrix [29]. In fact, lower levels of MMP1 protein, observed in our study, suggest a synthesis and deposition of Collagen I in the heart extracellular matrix. It is known that collagen I forms thicker and stiffer fibers, whereas the thinner reticular collagen III fibers are more flexible [30], which can explain the mechanical alterations observed with echocardiographic CS measurements in this study. Interestingly, we also found a strong correlation between echocardiographic CS and various parameters of myocardial fibrosis, suggesting the role of myocardial deformation as a surrogate marker of fibrosis. Similar findings have been previously described in hypertrophic cardiomyopathy and aortic stenosis [31] but not in CTRCD, although a correlation between CMR (T1 and T2 mapping) and subclinical histopathological changes in cardiotoxicity has been revealed [12].

In accordance with another study [12], we observed the presence of interstitial edema, which may contribute to changes in cardiac contractility. By examining AQP1 expression, we described, for the first time, a significant decrease in AQP1 expression in cardiac tissue treated with anthracycline. AQP1 has been linked to tumor development due to its ability to alter the expression of crucial cell cycle proteins, which appears to be associated with an increase in cell proliferation [32]. Moreover, it appears that AQP1 overexpression in breast cancer plays a role in chemotherapy sensitivity [33]. Actually, breast cancer patients with high levels of AQP1 expression who received the anthracycline treatment had better clinical outcomes as compared with those with low levels of AQP1 expression [34]. It is important to point out that CMR still remains the gold standard to detect myocardial edema after anthracycline treatment [35], but it is less accessible, and it is a time-consuming technique.

Anthracycline treatment can also be responsible for the loss of cardiomyocyte volume, as well as an expansion in the myocardial extracellular volume fraction due to edema and fibrosis. Herein, in WGA staining, we observed a tendency in the reduction in cellular size that was not statistically significant. Even though different studies in mice [36] and rats [12] evidenced severe myocardial edema assessed by CMR after several weeks of treatment, in another work, it has been described only at the beginning of the treatment but not after the completion of therapy [37]. Therefore, the analysis of the myocardial tissue composition could be useful in assessing the loss of cardiomyocyte volume more precisely.

In the current study, we also examined the behavior of miRNAs after anthracycline treatments because miRNAs have been reported as good biomarkers for predicting different types of cardiac damage. Among the differentially expressed miRNAs found in the Array, miR-122-5p was the most overexpressed in patients with CTRCD. MiR-122-5p has been reported to be related to heart fibrosis [38]. miR-122 levels have emerged as an early-warning biomarker of cardiovascular fibrosis [38]. A five-year longitudinal study (n = 79) on the long-term progression of diabetic cardiomyopathy using CMR and microArray analysis identified cardiac strain worsening and upregulation of miR122-5p, which was linked to matrix metalloproteinases (MMPs) and their regulators. Moreover, in db/db mice, miR122-5p overexpression was associated with diabetic cardiomyopathy and MMP-2 regulation [39]. A previous study proposed a relationship between the expression of miR-122-5p and increased levels of hs-TnT (ultrasensitivity cardiac Troponin T) in breast cancer patients treated with doxorubicin but not in patients treated with epirubicin [40]. Herein, we also demonstrated that miR-122-5p is significantly increased in the blood serum of rats treated with epirubicin. In silico analysis and correlation curves suggest that miR-122-5p may be involved in extracellular matrix remodeling that leads to fibrosis in accordance with previous findings [38]. Furthermore, miR-122-5p may regulate processes of senescence, autophagy, and apoptosis linked to the p53 pathway, as suggested by in silico analysis and previously demonstrated in various cancers [41,42,43]. The regulatory role of miR-122-5p in the p53 pathway seems beneficial for cancer treatments; however, it could potentially impact the behavior of cardiac cells as a contractile syncytium. When miR-122 is overexpressed in the heart of transgenic mice (n = 5), it causes cardiac dysfunction, morphological abnormalities, and cardiomyocyte apoptosis. Mechanistically, miR-122 promotes apoptosis by inhibiting the Hand2 transcription factor, which increases Drp1-mediated mitochondrial fission, potentially contributing to heart failure [44]. Considering all these data, further experiments are needed to elucidate the role of miR-122-5p in cardiotoxicity.

## 5. Conclusions

In conclusion, our study proposes that patients receiving anthracycline with a low baseline risk for cardiotoxicity (a population under-represented in previous studies) might show early changes in GLS and increased expression of miR-122-5p, which would allow the prediction of the future development of CTRCD. Using the animal model, we demonstrated the role of myocardial alterations and proposed miR-122-5p as a potential predictor for cardiac fibrosis and dysfunction.

### Limitations

The present study aimed to validate the ability of GLS and miRNAs for early detection of the development of CTRCD in a low-risk population and that cardiac strain correlates with the appearance of myocardial fibrosis. This study was performed in a single center, and the sample size is small due to the difficulty of recruiting patients with homogeneous criteria to be considered exclusively at low risk for cardiotoxicity. This fact limits the statistical power and is the reason why this study is descriptive, and conclusions should be taken with caution. A large-scale and multicenter study with a bigger sample size would be eagerly welcomed to confirm our findings. A small number of patients developed CTRCD, so our findings should be interpreted with caution. Furthermore, the study lacks a control group for possible confounding factors. Moreover, we did not perform CMR. According to the recently published cardio-oncology guidelines, CMR should be considered to assess cardiac function when transthoracic echocardiography is unavailable or not diagnostic [2]. Yet, CMR is a less accessible and time-consuming technique. In addition, strain analysis or parametric images, both necessary for a cardiotoxicity study, are still poorly available. If we had performed CMR, it would have added more strength to the study, but in our center, at the moment of this study, and as in most centers, CMR was not available (and did not have the necessary software) for research purposes in the field of cardiotoxicity.

## Figures and Tables

**Figure 1 biomedicines-13-00045-f001:**
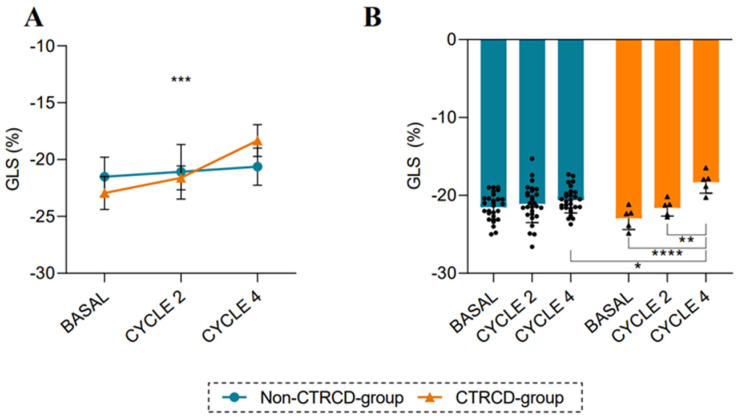
Changes in GLS during anthracycline treatment: (**A**,**B**) Evolution of GLS over time in CTRCD-group (n = 5) and non-CTRCD-group (n = 28). Values are presented as means ± SD. Repeated measures ANOVA and multiple comparisons were performed. (*), (**), (***) and (****) indicate significance at *p* < 0.05, *p* < 0.01, *p* < 0.001 and *p* < 0.0001.

**Figure 2 biomedicines-13-00045-f002:**
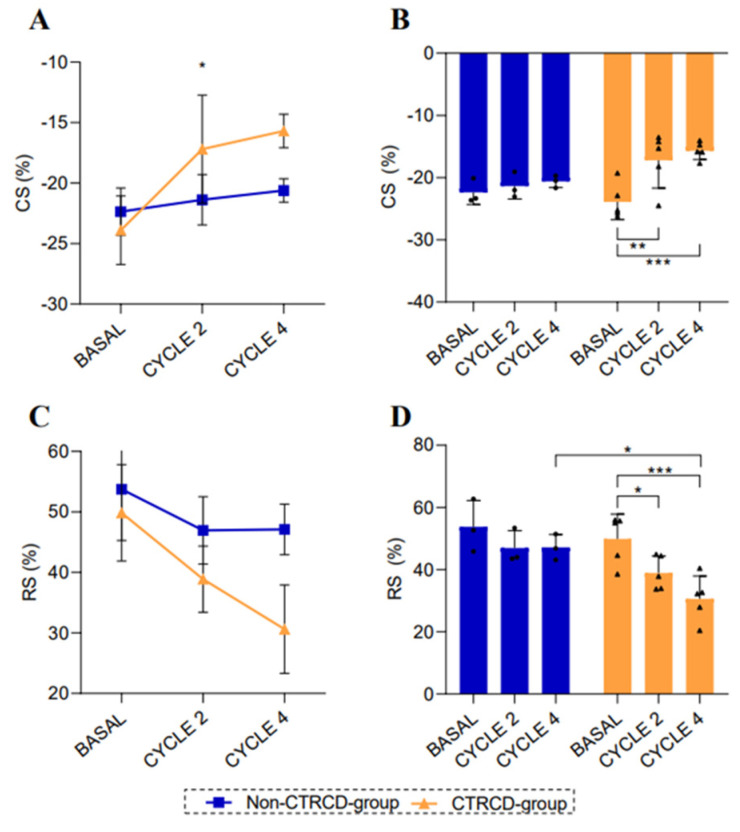
Evolution of echocardiographic parameters in the animal model: (n = 8). (**A**,**B**) Evolution of CS during anthracycline treatment in CTRCD-group (orange, n = 5) and non-CTRCD-group (blue, n = 3). (**C**,**D**) Evolution of RS during anthracycline treatment in CTRCD-group (blue) and non-CTRCD-group (orange). Values are presented as means ± SD. One-way ANOVA, repeated measures ANOVA, and Tukey’s multiple comparisons were performed. (*), (**), and (***) indicate significance at *p* < 0.05, *p* < 0.01 and *p* < 0.001, respectively. CS: circumferential strain; RS: radial strain; CTRCD: cancer-therapeutics-related cardiac dysfunction.

**Figure 3 biomedicines-13-00045-f003:**
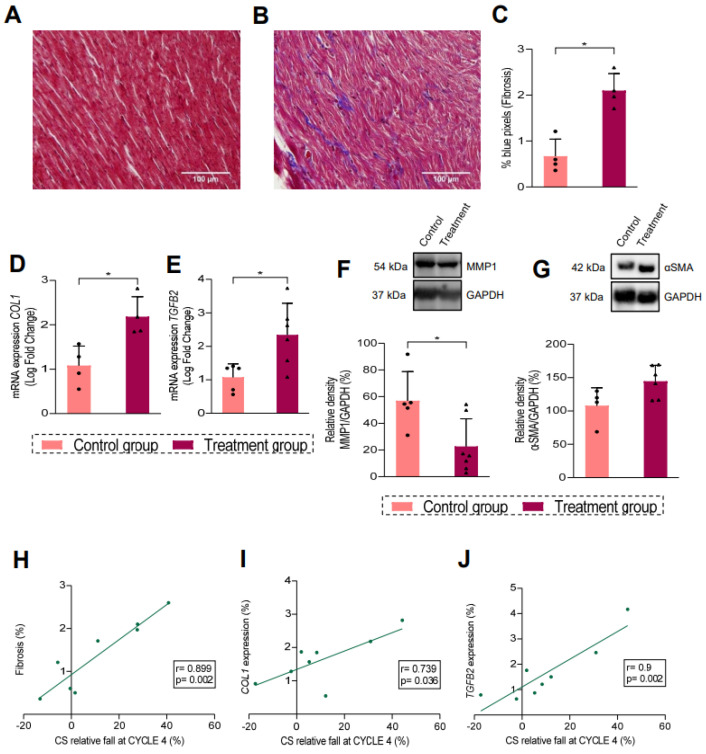
Study of myocardial fibrosis in the animal model. Representative images showing Masson’s Trichrome staining in control (**A**) and in treatment rat group (**B**). (**C**) Quantification of blue pixels in % in control (pink, n = 4) and treatment (burgundy, n= 4) group. (**D**,**E**) Relative mRNA expression of *COL1* gene (**D**) and *TGFB2* gene (**E**). Results of qRT-PCR for gene expression were normalized to endogenous control (*18S*). (**F**) Immunoblot of MMP1 and GAPDH image and quantification of relativity density of MMP1 normalized to GAPDH in bar graph (n = 5–7). (**G**) Immunoblot of α-SMA and GAPDH image and quantification of relativity density of α-SMA normalized to GAPDH in the bar graph (n = 4–6). (**H**) Correlation between fibrosis and CS relative fall at CYCLE 4. (**I**) Correlation between COL1 expression and CS relative fall at CYCLE 4. (**J**) Correlation between TGFB2 expression and CS relative fall at CYCLE 4. Fold-change > 1 means upregulation, while fold-change < 1 means downregulation. Data are shown as means ± SD. (*) indicates significance at *p* < 0.05. Pearson correlation test was performed (n = 8), with the correlation coefficient (r) (values given on the graphs). CS: circumferential strain; COL1: collagen 1; MMP1: metalloproteinase 1; αSMA, smooth muscle actin; TGFB2: transforming growth factor beta 2.

**Figure 4 biomedicines-13-00045-f004:**
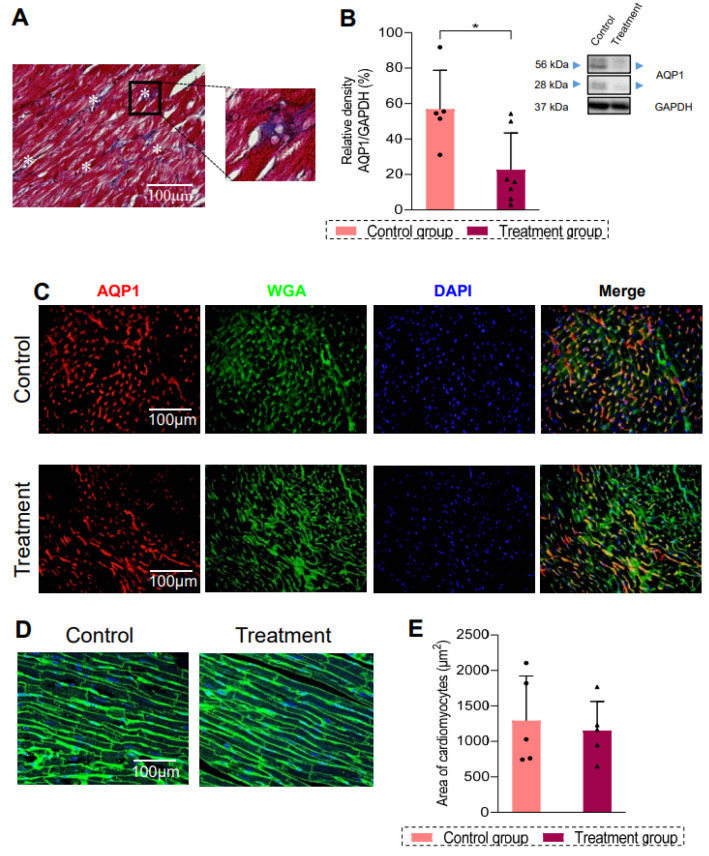
Study of edema in the animal model: (**A**) Representative image of Masson’s trichrome staining in a treatment group rat, where * indicates the interstitial vacuolar degeneration. (**B**) Quantification of relativity density of AQP1 normalized to GAPDH in control (pink, n = 4) and treatment (burgundy, n = 4) group and Immunoblot image of AQP1 (top: arrow indicate protein expression of dimer of AQP1; middle: arrow indicate protein expression of monomer of AQP1) and bottom: GAPDH. (**C**) Immunofluorescence staining of AQP1in control (upper panels) and treated rat hearts (down panels) (red, anti-AQP1; green, WGA; blue, DAPI), Scale bar = 100 µm. (**D**) Cardiomyocyte size assessed with WGA staining (green) in control and treated rat hearts at CYCLE 4 and quantification of cross-sectional cell size in µm^2^ (**E**). Scale bar = 100 µm. Data are shown as means ± SD. (*) indicates significance at *p* < 0.05. AQP1: aquaporin 1; SD: standard deviation.

**Figure 5 biomedicines-13-00045-f005:**
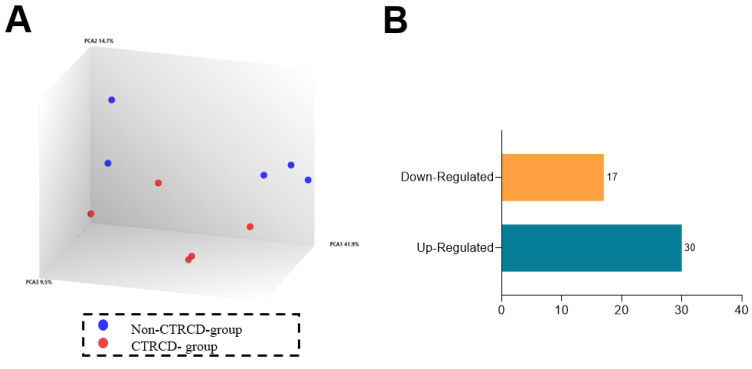
miRNAs differentially expressed between non-CTRCD-group and CTRCD-group: PCA of microRNAs (**A**) in Non-CTRCD-group (blue, *n* = 5) and CTRCD-group (red, *n* = 5) after CYCLE 4. Number of differently expressed genes (**B**): downregulated (yellow) and up-regulated (blue). Volcano plots (**C**) and hierarchical clustered (**D**) plots of −2.0 > fold-change > 2.0 value of microRNAs. In the heat map, non-CTRCD-group is in yellow and CTRCD-group in green. CTRCD: cancer-therapeutics-related cardiac dysfunction.

**Figure 6 biomedicines-13-00045-f006:**
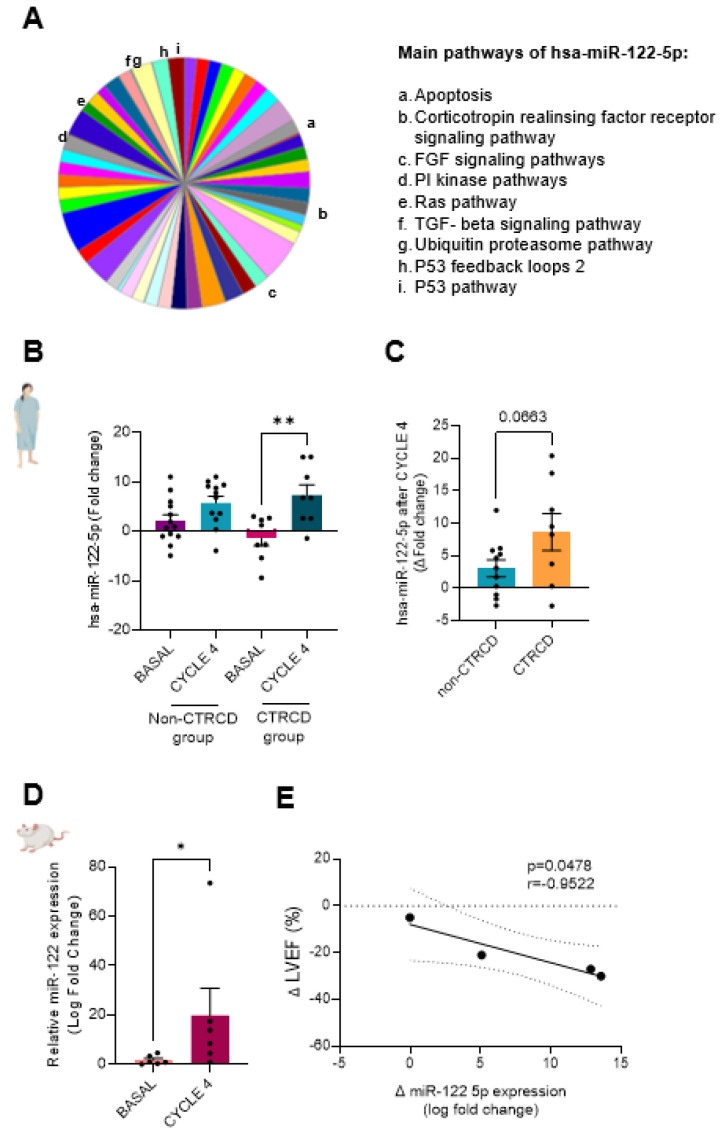
hsa-miR-122-5p in silico analysis and its validation in patients and rats: In silico study (**A**) of possible microRNAs regulating apoptotic and fibrotic genes. hsa-miR-122-5p validation in human serum (**B**) of non-CTRCD-group (basal n = 12, in purple; after CYCLE 4 n = 8, in pink) and CTRCD-group (basal n = 12 in turquoise; after CYCLE 4 n = 8 in blue) at basal time and after CYCLE 4. (**C**) hsa-miR-122-5p expression normalization to basal level in non-CTRCD-group (n = 12, blue) and CTRCD-group (n = 8, orange). hsa-miR-122-5p validation in treated rat serum (**D**) of treated rats (basal n = 6, in pink; after CYCLE 4 n = 6, in burgundy). PCA: principal component analysis; CTRCD: cancer-therapeutics-related cardiac dysfunction. (**E**) Simple linear regression of miR-122-5p and LVEF. LVEF: left ventricular ejection fraction. (*) and (**) indicate significance at *p* < 0.05 and *p* < 0.01 respectively.

**Table 1 biomedicines-13-00045-t001:** Baseline characteristics of the study cohort. Data are shown as means ± SD and as n (%). Student’s *t*-tests or chi^2^ test (Fisher test in case of n < 5) were performed. CTRCD: cancer-therapeutics-related cardiac dysfunction; GLS: global longitudinal strain; LVEF: left ventricular ejection fraction.

	Total (n = 33)	CTRCD-Group (n = 5)	Non-CTRCD-Group (n = 28)	*p* Value
Age (years)	52.8 ± 10.2	47.4 ± 7.5	53.7 ± 10.4	0.203
Arterial hypertension	8 (24.2%)	0 (0%)	8 (28.6%)	0.302
Dyslipidemia	4 (12.1%)	0 (0%)	4 (14.3%)	1.000
Diabetes Mellitus	2 (6.1%)	0 (%)	2 (7.1%)	1.000
Baseline GLS (%)	−21.7 ± 1.7	−22.9 ± 1.5	−21.5 ± 1.7	0.910
Baseline LVEF (%)	61.5 ± 4.9	62.9 ± 3.7	61.3 ± 5.1	0.510
Surgery	32 (96.9%)	5 (100%)	27 (96.4%)	1.000
Radiotherapy	31 (93.9%)	5 (100%)	26 (92.9%)	1.000
Left side	16 (48.5%)	4 (80%)	12 (42.9%)	0.333
Epirubicin	30 (90.9%)	5 (100%)	25 (89.3%)	1.000
360 mg/m^2^	22 (66.7%)	5 (100%)	17 (60.7%)	0.287

**Table 2 biomedicines-13-00045-t002:** Changes over time in GLS and LVEF. Data are shown as means ± SD. ANOVA tests were performed. GLS: global longitudinal strain; LVEF: left ventricular ejection fraction.

	Baseline	Cycle 2	Cycle 4	12 Month Follow-Up	*p* Value
GLS (%)	−21,73 ± 1,74	−21.16 ± 2.25	−20.28 ± 1.78	−20.77 ± 2.10	0.0266
CTRCD group	−22.94 ± 1.46	−21.60 ± 1.06	−18.32 ± 1.40	−20.86 ± 1.88	0.0159
Non-CTRCD group	−21.51 ± 1.72	−21.08 ± 2.40	−20.63 ± 1.63	−20.71 ± 2.27	0.0915
LVEF (%)	61.5 ± 4.9	60.72 ± 3.72	61.09 ± 4.92	58.8 ± 5.1	0.0948
CTRCD group	62.9 ± 3.7	60.6 ± 1.0	58.9 ± 5.2	57.3 ± 6.8	0.0476
Non-CTRCD group	61.3 ± 5.1	60.8 ± 4.0	61.5 ± 4.9	59.7 ± 3.7	0.5359

**Table 3 biomedicines-13-00045-t003:** Changes over time in LVEF, CS, and RS in the animal model. Data are shown as means ± SD. ANOVA tests were performed. LVEF: left ventricular ejection fraction; CS: Circumferential Strain; RS: Radial Strain.

	Baseline	CYCLE 2	CYCLE 4	*p* Value
LVEF (%)	90.0 ± 2.1	86.1 ± 5.4	74.9 ± 8.9	0.0002
CS	−23.31 ± 2.52	−18.76 ± 4.16	−17.53 ± 2.80	<0.0001
RS	51.30 ± 7.80	41.90 ± 6.57	36.81 ± 10.40	0.0081

**Table 4 biomedicines-13-00045-t004:** Expression of 21 microRNAs in patients at the final dose of epirubicin. Data are shown as means. Comparative analysis between non-CTRCD-group and in CTRCD-group was carried out using fold-change of over ± 2.0 with a *p*-value < 0.05. Patients’ age was applied as real covariate in the analysis. CTRCD: cancer-therapeutics-related to cardiac dysfunction.

Non-CTRCD-Group Average (log2)	CTRCD-Group Average (log2)	Fold Change	*p*-Value	microRNA
4.63	7.18	5.86	0.0158	hsa-miR-122-5p
2.89	3.98	2.13	0.0148	hsa-miR-3196
2.66	3.6	1.93	0.0361	hsa-miR-4516
0.34	1.21	1.83	0.0195	hsa-miR-6792-5p
4.45	5.11	1.58	0.0357	hsa-miR-6727-5p
2.18	2.82	1.56	0.0336	hsa-miR-6816-5p
0.15	0.76	1.52	0.0111	hsa-mir-663a
3.39	3.98	1.5	0.0484	hsa-miR-6869-5p
2.81	2.2	−1.52	0.0276	hsa-miR-4515
2.94	2.32	−1.54	0.0357	hsa-miR-4750-5p
4.48	3.86	−1.54	0.0263	hsa-mir-6742
0.65	0.02	−1.54	0.0038	hsa-miR-6080
5.18	4.54	−1.56	0.0061	hsa-miR-4428
1.4	0.73	−1.59	0.0035	hsa-miR-5004-5p
2.06	1.37	−1.61	0.0372	hsa-miR-5010-5p
2.89	2.15	−1.67	0.0028	hsa-miR-4436a
3.58	2.82	−1.69	0.0114	hsa-miR-564
0.78	0.01	−1.7	0.0336	hsa-miR-600
0.75	−0.11	−1.83	0.0052	hsa-miR-186-3p
3.49	2.6	−1.86	0.0044	hsa-miR-4283
3.62	2.49	−2.19	0.0036	hsa-miR-7151-3p

## Data Availability

The datasets used and/or analyzed during the current study are available from the corresponding author upon reasonable request.

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
