# Peer review of "Early Myocardial Strain Reduction and miR-122-5p Elevation Associated with Interstitial Fibrosis in Anthracycline-Induced Cardiotoxicity"

_biomedicines, 2024, doi:10.3390/biomedicines13010045_

Round 1
Reviewer 1 Report
Comments and Suggestions for Authors
Thank you for the opportunity to review this interesting article. The article covers the topical issue of cardiotoxicity of drugs used in oncology, since the number of patients at risk of cardiotoxicity increases every year, which is associated with successful cancer treatment. In the introduction, the authors describe the degrees of risk of cardiotoxicity. I would like to see a more specific and detailed classification of such risk depending on the presence of risk factors. It would also be interesting if the authors presented the odds ratios for the development of cancer-therapy related cardiac dysfunction in patients included in the study.
Author Response
Comments 1: Thank you for the opportunity to review this interesting article. The article covers the topical issue of cardiotoxicity of drugs used in oncology, since the number of patients at risk of cardiotoxicity increases every year, which is associated with successful cancer treatment. In the introduction, the authors describe the degrees of risk of cardiotoxicity. I would like to see a more specific and detailed classification of such risk depending on the presence of risk factors.
Response 1: We would like to thank the Reviewer 1 for his/her comments regarding our study and for recognizing the growing clinical significance of cardiotoxicity. In this revised manuscript we have added more details about classification of the risk for cardiotoxicity as highlighted in the introduction paragraph 1 line 57.
Comments 2: It would also be interesting if the authors presented the odds ratios for the development of cancer-therapy related cardiac dysfunction in patients included in the study.
Response 2: Thank you for your wise suggestion. We agree that odds ratio analysis would be very useful to implement the score for low-risk patients. However, as this valuable reviewer knows this analysis is useful for case-control studies, and the sample in studies involving patients must be large enough to detect a clinically significant difference. To assess this statistical test, each cell in the contingency table should have at least 5 observations for a reliable estimate. Our study has a limitation with a small cohort size, which is at the statistical limit. Moreover, we lack of a control group of breast cancer patients with no administration of anthracyclines to assess the risk of developing cardiac dysfunction after chemotherapy treatment. We will take in consideration your suggestion for next study with a larger cohort.

Reviewer 2 Report
Comments and Suggestions for Authors
The manuscript investigates an under-represented population in cardiotoxicity studies, focusing on low-risk patients treated with anthracyclines. It successfully highlights the relationship between myocardial strain reduction, fibrosis, and miR-122-5p, providing valuable insights for early detection and mechanistic understanding of cardiotoxicity. The use of both clinical and animal models strengthens the study. miR-122-5p provides a potential diagnostic tool for anthracycline-induced cardiotoxicity. However, there are some points that need to be improved:
- The small cohort size (n=33) limits the statistical power and generalizability. This limitation is acknowledged but needs further emphasis in the discussion.
- The study lacks of control for confounding factors such as variations in patient treatment or comorbidities.
- The absence of cardiac MRI, despite its acknowledged value in detecting myocardial edema and fibrosis, reduces the study's comprehensiveness. Information from MRI would add more strength in the study.
- miR-122-5p is not yet validated in clinical patients. Thus, its use is still controversial.
- Also, reduce pathophysiological mechanisms from Introduction and write them in the discussion.
- In methods, details about miRNA array processing should include reproducibility checks or replicate analyses.
- In the discussion, I suggest to compare your results with larger datasets or similar studies.
Author Response
Comments 1: The manuscript investigates an under-represented population in cardiotoxicity studies, focusing on low-risk patients treated with anthracyclines. It successfully highlights the relationship between myocardial strain reduction, fibrosis, and miR-122-5p, providing valuable insights for early detection and mechanistic understanding of cardiotoxicity. The use of both clinical and animal models strengthens the study. miR-122-5p provides a potential diagnostic tool for anthracycline-induced cardiotoxicity. However, there are some points that need to be improved:
Response 1: Thank you very much for your valuable’s comments and suggestions. Indeed, they helps us to improve substantially this revised version as highlighted through the text.
Comments 2: The small cohort size (n=33) limits the statistical power and generalizability. This limitation is acknowledged but needs further emphasis in the discussion.
Response 2: Thank you for your comment. It was very difficult to recruit patients with homogeneous criteria to be considered only at low risk for cardiotoxicity, resulting in a small cohort size which limited the possibility of performing advanced statistical studies. This is the reason why we only performed a descriptive study with our small clinical cohort, and conclusions should be taken with caution and as a proof-of-concept study. Future multicentre studies with a bigger sample size, focused on this profile of patients, would be very interesting. Now the small cohort number is emphasized in page 16 in line 499, and in the limitations section in page 18 line 598, as highlighted.
Comments 3: The study lacks of control for confounding factors such as variations in patient treatment or comorbidities.
Response 3: We wish to thank you for your appreciation. You are correct, the study does not control for confounding factors such as variations in patient treatment or comorbidities. We are aware of this limitation since these factors can influence the outcomes and bias the results. Here, we have provided, in table 1, information about cardiovascular risk factors that gave a basic idea regarding patient’s profile of this small cohort. Even if these results should be interpreted with caution, they do offer a basis for future research. Also, the age of the patients (52.8 years) makes it difficult to select patients without cardiovascular risk factors. As stated above, a larger and more diverse sample size will ensure more evenly distributed variations in treatment and comorbidities. Now highlighted in Limitations section line 602, page 18.
Comments 4: The absence of cardiac MRI, despite its acknowledged value in detecting myocardial edema and fibrosis, reduces the study's comprehensiveness. Information from MRI would add more strength in the study.
Response 4: Thanks again for your appreciation. We are aware that MRI is the best imaging technique for studying changes in cardiac contractility and edema. As noted in the discussion, performing CMR would have added greater strength to our study. However, at the time of the study, our centre did not have access to CMR (nor the necessary software) for research purposes in the field of cardiotoxicity. Actually, our results are in line with a recent study by Fabian Voß and colleagues, who used CMR in a cohort of breast cancer patients with low risk of cardiotoxicity, finding an increased number of patients (54,2%) with a reduction in LVEF after 12 months of the treatment, and strain fall. (Voß F., et al. Anthracycline therapy induces an early decline of cardiac contractility in low-risk patients with breast cancer. Cardio-Oncology 2024, 10, (1) 43, doi:10.1186/S40959-024-00244-Y). We added new sentences regarding this data in line 507, page 16, and in line 608, page 18.
Comments 5: miR-122-5p is not yet validated in clinical patients. Thus, its use is still controversial.
Response 5: Thank you for the observation. Our data propose measuring miR-122-5p in the peripheral blood of patients undergoing chemotherapy, alongside existing markers, to strengthen conclusions about the increased risk of cardiotoxicity. In addition, while miR-122 has not yet been clinically associated with cardiotoxicity, numerous publications have demonstrated its relationship with conditions such as hypertension, atherosclerosis, atrial fibrillation, acute myocardial infarction, and heart failure (Liu et al. Roles of MicroRNA-122 in Cardiovascular Fibrosis and Related Diseases. Cardiovasc Toxicol. 2020 Oct;20(5):463-473. doi: 10.1007/s12012-020-09603-4.), now discussed in line 563, page 17. For example, Vogel et al. found elevated miR-122 levels in patients with systolic heart failure, although they did not characterize its specific role in cardiac function. It will be interesting to validate miR-122-5p for clinical use by rigorous clinical trials conducted in large cohorts, now discussed in line 579, page 18 (Vogel B et all. Multivariate miRNA signatures as biomarkers for non-ischaemic systolic heart failure. Eur Heart J. 2013 Sep;34(36):2812-22. doi: 10.1093/eurheartj/eht256. Epub 2013 Jul 17. PMID: 23864135.).
Comments 6: Also, reduce pathophysiological mechanisms from Introduction and write them in the discussion.
Response 6: As suggested, we moved pathological mechanisms from the introduction to discussion section, as highlighted in page 16, line 488. We have also summarized the mechanisms of action of anthracyclines in the introduction.
Comments 7: In methods, details about miRNA array processing should include reproducibility checks or replicate analyses.
Response 7: Following your wise suggestion, we have introduced a sentence about miRNA processing, as highlighted in line 204 page 5 in method section and line 227 page 5.
In the current study, miRNA array was performed in representative samples of patients (n = 5). Subsequently and after in silico analysis the most differentially expressed microRNAs were validated through RT-qPCR in more patients (n = 20). Each sample was analysed in triplicate, and samples presenting a Ct value >35 and a Ct confidence value ≤0.6 were excluded.
Comments 8: In the discussion, I suggest to compare your results with larger datasets or similar studies.
Response 8: Thank you for the suggestion. Now we have discussed results from a recent work by Fabian Voß (Voß F., et al. Anthracycline therapy induces an early decline of cardiac contractility in low-risk patients with breast cancer. Cardio-Oncology 2024, 10, (1) 43, doi:10.1186/S40959-024-00244-Y), who used CMR to assess cardiac function in larger cohort of breast cancer patients (n = 59) with low risk of cardiotoxicity. The study described that 33 of patients (54,2%) developed significant reduction in LVEF, together with longitudinal strain fall after 12 months of the treatment as highlighted in page 16, line 507. Perhaps, using CMR, more sensitive imaging technique allowed such increase in the detection of cardiotoxicity in this cohort.

Round 2
Reviewer 1 Report
Comments and Suggestions for Authors
I suggest accepting the article as it is
Reviewer 2 Report
Comments and Suggestions for Authors
Authors have answered all comments sufficiently.